# Environmental Regulation, Corporate Social Responsibility (CSR) Disclosure and Enterprise Green Innovation: Evidence from Listed Companies in China

**DOI:** 10.3390/ijerph192214771

**Published:** 2022-11-10

**Authors:** Xiumei Xu, Ruolan Jing, Feifei Lu

**Affiliations:** School of Economics and Management, Qingdao Agricultural University, Qingdao 266100, China

**Keywords:** CSR disclosure, green operation management, emerging economies

## Abstract

The resource and environmental constraints on China’s economic development have become more prominent; thus there is an urgent need for enterprises to achieve green innovation transformation to promote high-quality economic development. We obtained data on 655 on Chinese A-share companies listed on the Shanghai and Shenzhen Stock Exchanges from 2010 to 2020, a total of 7205 samples, and explored the influencing mechanism of environmental regulation on corporate green innovation and the moderating mechanism of CSR disclosure by constructing a nonlinear fixed-effect regression model. The results showed: (1) the overall level of green innovation of listed companies is low, and the relationship between environmental regulation and enterprise green innovation presents the U-shaped characteristic of changing from cost effect to innovation compensation effect; (2) non-state-owned enterprises have less tolerance and more sensitive response to environmental regulation than state-owned enterprises; (3) social responsibility information disclosure has a positive regulatory effect on environmental regulation and enterprise green innovation, and non-state-owned enterprises are more significant. It provides references for the government to adjust the intensity of environmental regulation, and, meanwhile, for enterprises to improve the level of environmental protection and the CSR disclosure, and enhance the green innovation ability of enterprises in emerging market.

## 1. Introduction

In recent years, China’s economic growth rate has been among the highest in the emerging economies, creating remarkable achievements in economic development, but the ensuing problems of environmental pollution, resource depletion and ecological damage have become more and more prominent [1,2]. Steadily promoting high-quality economic development urgently requires building a green, low-carbon and circular development economic system in which green innovation plays leading and supporting roles. As an emerging technology model, green innovation takes green development as its core pursuit. It can optimize resource allocation, reduce natural resource consumption and ecological damage, coordinate the relationship between economic development and resources and the environment and enable high-quality economic development. As an important support for economic development, enterprises are also the main seekers of resources, the main body of environmental pollution and the important promoters of green transformation and green innovation. exploring the influencing factors of enterprise green innovation has important theoretical value and practical significance for stimulating the enthusiasm for corporate green innovation, accelerating the transformation of corporate green innovation achievements and promoting high-quality development.

However, due to the huge cost of environmental management, enterprises lack the motivation to pursue active green innovation. There are many factors affecting the green innovation of enterprises. Scholars mainly study from two aspects: internal governance and external pressure. External pressure mainly includes regulatory pressure, normative pressure and imitation pressure [3]. As a regulatory pressure, environmental regulation transforms environmental protection needs into strict policies by providing normative content and is considered an important measurement tool for achieving green development in enterprises, but whether there is a positive compensation effect or a negative offset effect is not agreed upon [2,3,4].

Internal governance mainly emphasizes corporate management’s values [5] and social responsibility cognition [6]. Faced with the same external institutional pressure, enterprises have different degrees of green innovation that to a certain extent depends on the attitude of actively seeking social responsibility [7]. Actively assuming corporate social responsibility and disclosing accountability information helps enterprises to obtain legitimacy, win public support, bring more financing opportunities, reduce the pressure on innovation funds, provide a steady stream of funds for continuous innovation and finally realize the sustainable development. During the course of enterprises’ actively seeking their own economic interests and assuming social responsibility, enterprise social responsibility disclosure will affect the relationship between environmental regulation and enterprise green innovation. As two important driving forces to promote the green innovation of enterprises, what are the relationships of environmental regulation, social responsibility information disclosure and enterprise green innovation? Is there an interaction between environmental regulation and social responsibility information disclosure? are the focus of this paper. This paper analyzes the impact of government environmental regulation on enterprise green innovation and deeply discusses the regulatory role of social responsibility information disclosure in the relationship between environmental regulation and enterprise green innovation.

Under the background of market economy, facing the positive externalities of knowledge spillover and negative externalities of environmental pollution presented by green innovation, it is difficult for enterprises to spontaneously invest in green innovation due to the comprehensive consideration of their own interests. In the institutional environment, institutional theory suggests that the source drivers of green innovation for enterprise come from institutional pressures, including those of government environmental regulation. How environmental regulation affects enterprise green innovation has been widely debated by academics in recent years, and the relevant findings are mainly centered on Porter’s hypothesis. Berrone et al. [8] argued that policy pressure from government environmental regulation is the main driver of enterprise green innovation and his findings strongly support the Porter hypothesis. After the government imposes appropriate environmental regulation, enterprises may choose strategies that are consistent with environmental regulatory standards based on their relative competitive advantages in development, initiate green innovation to counteract system compliance costs and form innovation compensation [9,10,11]. For example, Ambec et al. [12] showed that market-incentivized environmental regulation policies form a complementary relationship with the company’s internal governance mechanism while putting pressure on enterprises, thereby mobilizing the enthusiasm of enterprises for green innovation. Paola et al. [13] found that enterprises were more inclined to apply for green patents after the implementation of new environmental regulations. Jinjarak et al. [14] found that environmental regulation greatly constrains the environmental degradation of enterprise and plays an important role in environmental protection. Kesidou et al. [15] showed a positive causal effect between the severity of environmental regulations and the green business performance of enterprises. Using data on the green patents of listed enterprises as a sample, Kraus et al. [16] found that a carbon trading pilot as an environmental regulation policy induced green innovation by enterprise in the pilot region. Mavragani et al. [17] examined the extent to which the openness of a market economy and the quality of the institution affect environmental performance, showing that the environmental performance index is positively correlated to institutional indicators. Thanh et al. [4] found that more comprehensive environmental management enables organizations to cope with the institutional pressures they face and thus improve environmental performance. Zhaoqiang et al. [18] applied a PSM–DID model to test the influence of environmental regulation on enterprise green innovation, indicating that the NEPL has significantly promoted the green innovation of heavily polluting enterprises; compared with non-state-owned enterprises, the NEPL plays a more significant role in promoting the green innovation of state-owned enterprises; and environmental responsibility plays a mediating role in the impact of environmental regulation on enterprise green innovation. Fan Wang et al. [19] showed that a higher intensity of environmental regulations is more beneficial for incentivizing enterprises to implement green innovation, the tenure length of officials plays an inverted U-shaped role in regulating the impact of environmental regulations on enterprise green innovation. Jinyong et al. [20] found that both environmental uncertainty and environmental regulation promote enterprises’ green technological innovation, and environmental regulation has positive moderating effects on the relationship between environmental uncertainty and enterprises green technological innovation. Wu Bao et al. [21] used a sample of 4924 private Chinese companies and indicated that both formal and informal regulation pressures have a positive effect on green innovation. However, some scholars were opposed to Porter’s hypothesis, arguing that environmental regulation is essentially an additional cost imposed by the government on enterprises [22,23], which to some extent discourages enterprise’ willingness and ability to innovate green and produces a crowding-out effect on green innovation investment. For example, based on a cross-country perspective, Allen et al. [24] found that environmental regulatory volatility reduces green innovation output by increasing uncertainty and unpredictable risk for enterprises and investors, significantly inhibiting their willingness to undertake green innovation activities. Petroni et al. [25] pointed out that enterprises are under heavy financial pressure due to environmental regulations, for which they have to take measures such as production and work stoppages, thus reducing the funds available for green innovation. Leeuwen et al. [26] empirically showed that environmental regulations reduce the productivity of manufacturing enterprises and are detrimental to their green innovation. Li et al. [27] constructed a theoretical model of GTIB in construction enterprises to analyze the mechanism of action of the factors influencing the GTIB of construction enterprises, showing that direct government investment has the greatest impact on their GTIB and has made a positive contribution and that the role of environmental regulation on the GTIB is nonlinear. Mingyue et al. [28] found that there is an inverted U-shaped relationship between environmental regulation and enterprise green technological innovation, and external financing constraints will reduce the impact of environmental regulation on enterprise green technological innovation. Thus, numerous scholars have gradually realized that there may be nonlinear characteristics of the impact of environmental regulation on green innovation; that is, there is a threshold effect between environmental regulation and green innovation [29,30,31]. When the government regulation of environmental pollution is weak and enterprises face relatively weak penalties, enterprise management has little incentive to engage in environmental governance and has a fluke mentality to avoid green innovation, whereas strict environmental regulation facilitates enterprise funding for greener, environmentally friendly industries and increases spending on green innovation [32,33,34]. In summary, although there are valuable results in the literature on environmental regulation and green innovation, no consensus has been reached in the academic community so far. In order to achieve green innovation incentives for enterprises, it has become a challenge for the government to optimally adjust and choose the appropriate level of environmental regulation enforcement according to their actual situation and differences.

Corporate social responsibility (CSR) disclosure is a vital bridge for firms to communicate with their stakeholders and achieve legitimacy. Studies have shown that enterprises that can take into account both social and environmental issues are more able to persist in developing green innovation in the long run [35]. In recent years, the lack of enterprise social responsibility awareness has triggered strong demands for social responsibility information disclosure from all walks of life. As a source of information for listed companies, enterprise social responsibility information disclosure directly affects the company information obtained by investors and is reflected in innovation investment risks. As such, can enterprises’ disclosure of social responsibility information motivate them to carry out green innovation? This question has aroused widespread concern in the academic community. Based on the theory of information asymmetry and signal transmission, some scholars believe that managers will disclose social responsibility information out of the motivation of ethics and corporate strategic development. They reduce information asymmetry by transmitting the information of the company’s sustainable development capability to the stakeholders [36], narrow the information gap between investors and enterprises [37,38], reduce the cost of capital, improve the financing environment of enterprises and the reputation in the capital market [39] and create a good image to influence the green innovation investment of enterprises [40]. For example, Zhao Li et al. [41] explored the influence of different public participation constraints on green technological innovation and examined the moderating role of environmental regulatory enforcement by dividing public participation constraints into news media, community resident and ENGO constraints, finding that public participation has become an important force in the promotion of enterprises’ green technological innovation. Alan et al. [42] found that the disclosure of social responsibility information by listed companies can reduce information asymmetry, reduce forecasting and liquidity risks and further reduce the cost of equity capital. Berchicci et al. [43] pointed out that it will create more opportunities to gain social attention and receive environmental subsidies from the relevant government departments if an enterprise actively fulfills its social responsibilities such as environmental protection, thus greatly increasing the intensity of the enterprise’s environmental investment and promoting green technology and product updates. Doshi et al. [44] found that social responsibility information disclosure can reduce the degree of information asymmetry of companies, avoid penalties such as investor risk premiums or weaken investment and reduce the financing cost of enterprises. Dhaliwal et al. [45] found that enterprises that disclose social responsibility information have small errors in their earnings forecasts. Wang et al. [46] showed that social responsibility information disclosure drives the green transformation of enterprises by strengthening the legitimate motivation of enterprises. Belmonte Urena et al. [47] used bibliometric techniques and evaluated the contribution of current academic research to the advancement of sustainable development agenda as expressed in the UN sustainable development goals targets. However, other studies have come to the opposite conclusion. Based on principal–agent theory, scholars believe that managers may conduct social responsibility disclosure out of self-interest motives [48,49]. They try to divert the attention of stakeholders to cover up the bad behavior of enterprises. This kind of insubstantial information disclosure increases the financial cost of the enterprise, reduces the economic benefits to the enterprises and is not conducive to the quantity and quality of enterprises’ R&D investment [50,51]. For example, Atif et al. [52] showed that social responsibility contracts are often awarded by enterprises that focus on environmental protection and other stakeholder rights. Zhi et al. [53] found that the management of enterprises with lower social responsibility ratings is more likely to manipulate profits to achieve personal interests, thereby ignoring the innovation investment of enterprises.

From the existing literature achievements, scholars at home and abroad have extensively discussed the relationship between environmental regulation and green innovation, but a consensus has not been reached, and very few people pay close attention to the impact of environmental regulation in the lag period. There are also a few studies on the relationship between social responsibility information disclosure and enterprise innovation, and the existing studies discuss environmental regulation and social responsibility information disclosure separately, lacking a comprehensive exploration of the potential relationship between the two. In fact, enterprises reduce resource consumption and environmental pollution damage through green technology innovation, which is not only manifested in the obvious external costs to enterprises under the pressure of government environmental regulations but also were reflected in the active innovation behaviors of enterprises actively disclosing social responsibility information to strive for stakeholder value recognition or market demand opportunities. Therefore, it is necessary to examine the synergistic effect of environmental regulation on the incentives for green innovation from the perspective of internal responsible entities. 

This study takes green technology innovation theory, environmental regulation theory and CSR disclosure theory as the theoretical basis, uses a nonlinear fixed-effects model and tests the relationships among environmental regulation, social responsibility information disclosure and enterprise green innovation, which answers the following important questions:Can environmental regulation in the current period and in the lagging period promote enterprises’ green innovation?Is the relationship between environmental regulation and green innovation different between state-owned and non-state-owned enterprises?Does CSR disclosure have a certain regulatory effect on environmental regulation and enterprise green innovation?Does CSR disclosure have different effects in state-owned and non-state-owned enterprises?

The possible contributions and innovation of this paper are as follows. On the one hand, it provides evidence for the debate on whether the relationship between environmental regulation and enterprise green innovation is linear or nonlinear. At present, most research on the economic consequences of environmental regulation focus on regional [30] and industry [11]. This paper conducts an in-depth investigation from the perspective of enterprises and finds that weak environmental regulation has a cost effect on enterprise green innovation, while strict environmental regulation has an innovation compensation effect on enterprise green innovation. This study also enriches Porter’s hypothesis and environmental regulation and institutional pressure theories. On the other hand, this paper enriches Schumpeter’s innovation theory by taking green innovation as the breakthrough point. From the perspective of internal and external incentives, this paper attempts to use the resource-based theory and competitive advantage theory to interpret the mechanism of CSR disclosure on environmental regulation and enterprise green innovation. We verify that the CSR disclosure can significantly strengthen the relationship between environmental regulation and corporate green innovation, which not only can deepen the research on the mechanism of social responsibility information disclosure and the pre-drivers of corporate green innovation but also can expand the research perspective of stakeholder theory.

The main structure of this paper is as follows: The second chapter states the research hypotheses; the third chapter explains the research design process, including sample and data selection, variable definition and model building; the fourth chapter describes an empirical test of the relationship between environmental regulation, CSR disclosure and green innovation; the fifth chapter gives the robustness test process and results; the sixth chapter further discusses and explains the similarities and differences between this article and related research and the seventh chapter summarizes the full text and proposes ways to optimize CSR disclosure including suggestions for formulating environmental regulations to improve green innovation in enterprises.

## 2. Theoretical Analysis and Hypothesis Development

### 2.1. Environmental Regulation and Enterprise Green Innovation

Enterprises are the main creators of social wealth and the main body of negative externalities such as environmental pollution. Since the environment has the attributes of public goods [54], enterprises themselves do not need to pay for their resource requirements and environmental pollution. In addition, enterprises need to invest a great deal of manpower, material and financial resources in green innovation. If there is a lack of government environmental control, enterprises will not consume many resources for environmental protection to carry out green innovation. 

The level of enterprise green innovation varies with the intensity of government environmental regulation. In the period of weak environmental regulation, the increasing of environmental regulation intensity will aggravate the penalties for enterprises’ pollution discharge and increase the cost of pollution control [55,56]. In order to meet the government’s environmental protection requirements and avoid paying pollution penalties, enterprises tend to invest funds in the field of environmental governance [57] to control pollutant emissions and improve pollution control. However, green innovation requires high R&D costs. In order to meet their own economic interests, profit-oriented enterprises will increase capital investment in the production field to compensate for pollution control costs, thus squeezing out the capital investment in green innovation and reducing the development of green innovation activities [58]. In the period of strict environmental regulation, the cost of pollution control and punishment is higher than the economic benefits of green innovation. If enterprises choose to suspend production or relocate, they will not only lose the original market competitiveness, but also the cost of previous input will become sunk costs [59,60]. Under the trade-off, enterprises will be more inclined to carry out green innovation locally. After gradually adapting to the government’s environmental regulation, the green innovation R&D funds are now constantly increasing. With the enhancement of innovation ability, the production efficiency of enterprises will continuously improve, the production costs will gradually decrease, the benefits brought by green innovation will gradually offset the costs of environmental regulation and the green innovation output of enterprises will continuously increase. Based on this, the following hypotheses are proposed:

**H1a.** 
*There is a U-shaped curve relationship between the intensity of environmental regulation and enterprises’ green innovation; that is, with the increasing of the intensity of environmental regulation, the output of enterprises’ green innovation first decreases and then increases.*


**H1b.** *Considering the time lag of policy implementation, environmental regulation with a lag period has a more significant impact on green innovation*.

### 2.2. The Regulatory Effect of CSR Disclosure

The development of enterprise green innovation is not only affected by the pressure of external environmental policies but also driven by their own pursuit of economic interests. Social responsibility information disclosure is a long-term development strategy for enterprises to coordinate their own economic and social benefits. The relationship between environmental regulation and enterprise green innovation depends to varying degrees on the quality of enterprise social responsibility information disclosure.

On the one hand, environmental regulation will increase enterprises’ institutional compliance costs and the investment used for environmental governance [61], which makes enterprises bear heavy financial pressure. From the perspective of stakeholders, high-quality social responsibility information disclosure can convey to stakeholders a signal that the enterprise has sustainable development capabilities. Stakeholders are willing to give more value returns to enterprises based on their self-perception, thereby making up for the cost of environmental protection paid by enterprises to meet the requirements of government environmental control and alleviate the tension of the shortage of green innovation funds [62]. It is mainly manifested in the following three aspects. First, according to the basic resource theory, disclosing high-quality environmental and other social responsibility information is not only a cost input but also an investment behavior. It can reflect the implementation of the government’s environmental control measures by enterprises, help to obtain the legitimacy of the government [63,64,65,66] and obtain tax relief and financial subsidies from the government. Second, according to the theory of signal transmission and information asymmetry, the higher the quality of corporate social responsibility information disclosure, the more investors can accurately understand its internal management level and external business environment and reduce information asymmetry [67] and investment assessment risk. At the same time, it is helpful for enterprises to avoid penalties such as risk premium [68], and it can attract high-quality investors to increase investment and alleviate financing constraints. As a result, a virtuous circle of interests is formed between enterprises and investors that introduces long-term financial guarantees for the green R&D activities of enterprises. Third, according to Porter’s theory of competitive advantage, enterprises with high-quality social responsibility information disclosure tend to pay more attention to their social responsibility image. They will meet consumers’ expectations by reducing environmental damage in production and improving production techniques, thereby improving customer satisfaction and enterprise reputation, gaining higher market competitiveness and social benefits and meeting and subsidizing enterprise green R&D funds.

On the other hand, social responsibility disclosure will affect the performance of enterprises in environmental protection. The management of enterprises with high-quality social responsibility information disclosure often has a high sense of social responsibility and environmental protection awareness. They are more likely to identify potential opportunities and core competitive advantages brought by meeting the government’s environmental regulation requirements. They are also more willing to regard government environmental regulation as the potential external supervision of enterprises and turn the environmental protection pressure brought by environmental regulation into the driving force that can improve the operating efficiency and performance of the enterprise. In this way, it can make a positive response to the operating conditions and environmental problems of the enterprise, improve the utilization rate of its own resources by allocating of internal resources rationally, increase innovation investment in green technology and accelerate the transformation of green innovation achievements. Based on this, the following hypothesis is proposed:

**H2.** *CSR disclosure has a positive regulatory effect on the relationship between environmental regulation and enterprise green innovation. That is, high-quality social responsibility information disclosure can improve the overall U-shaped curve between environmental regulation and enterprise green innovation*.

### 2.3. Analysis of the Green Innovation Behavior of Enterprises with Heterogeneous Property Rights

Enterprises with heterogeneous property rights have different green innovation behaviors, and non-state-owned enterprises have higher R&D investment willingness than state-owned enterprises. Non-state-owned enterprises have a clear property rights structure of responsibilities and interests, there are no complicated principal–agent problems, and their capital owners tend to pursue the maximization of capital utilization efficiency. Under the pressure of environmental regulation and in the face of fierce market competition, non-state-owned enterprises in a relatively weak market position are more committed to improving production and innovation efficiency. They are willing to put more funds and energy into green innovation [69], which can gain substantial competitive advantage. State-owned enterprises often pursue the coexistence of political stability and economic benefits, while green innovation has uncertain returns and investment risks. Therefore, facing the implementation of environmental regulation policies, state-owned enterprises lack competitiveness in the input and output of green innovation. 

According to the expectation theory, good social responsibility information disclosure can meet the expectations of stakeholders, gain legitimacy and profit from it [70]. For stakeholders, whether state-owned enterprises actively fulfill their social responsibilities is related to the country’s stability and people’s well-being and the economic lifeline of the country, so it has a strong rigidity. In contrast, non-state-owned enterprises have weaker constraints on fulfilling their social responsibilities. According to the signal theory, once non-state-owned enterprises actively undertake social responsibilities, they will send a signal of their good development prospects to the outside world. Stakeholders will be willing to give greater economic returns to make up for the cost of enterprise environmental regulation and increase the funds needed for green innovation. Based on this, the following hypothesis is proposed:

**H3.** *Environmental regulation has a greater impact on the green innovation of non-state-owned enterprises, and non-state-owned enterprises that disclose high-quality social responsibility information have a more significant regulatory effect on environmental regulation and green innovation*.

The research framework and hypothesis relationship is shown in Figure 1. First, we propose H1a and H1b that there is a U-curve relationship between environmental regulation and corporate green innovation in the current period and lagged periods, respectively, for the full sample. Based on H1a, we propose H2 that CSR disclosure positively moderates the U-curve relationship between environmental regulation and corporate green innovation. Further, dividing the full sample into state-owned enterprises and non-state-owned enterprises according to different property rights natures, we propose H3 that non-state-owned enterprises are more sensitive to green innovation than state-owned enterprises facing the same intensity of environmental regulations, and non-state-owned enterprises with high-quality social responsibility information disclosure have a more significant regulatory effect on environmental regulation and green innovation.

## 3. Research Design

### 3.1. Sample and Data Sources

We select Chinese A-share listed companies on the Shanghai and Shenzhen Stock Exchanges from 2010 to 2020 as the initial sample. The main reasons for the selection of the time range are as follows. On the one hand, the data for measuring CSR based on the perspective of green innovation in this paper mainly come from the Hexun.com CSR rating database, which has continuously rated A-share listed companies since 2010; on the other hand, it lies in the fact that the concept of green development has been gradually strengthened in China since 2012. In order to ensure the validity of the data, we conduct the following screening: (1) We exclude companies that are ST (ST is the abbreviation of special treatment, which refers to the stock of a domestic listed company that has suffered losses in operation for two consecutive years and has been warned of the risk of delisting) or *ST (*ST refers to a stock that is about to be delisted, when a listed company has been operating for three consecutive years of losses or less than three years but has serious financial losses) during the sample period. (2) We exclude companies with missing or discontinuous data, those specializing in environmental protection business and those with outliers in financial indicators. (3) We supplement some of the missing financial indicators by reviewing the company’s annual report. (4) To control for the effects of the extreme values, we winsorize at the upper and lower 1% for all continuous variables. Finally, we obtain data on 655 listed companies in China from 2010 to 2020, a total of 7205 samples. Among them, the industry division refers to the 2012 edition of the SEC industry classification standards, involving three major industries, such as mining, manufacturing and electricity, heat, gas and water production and supply. The data sources for the study sample include: the data on the number of green patent applications and the total number of patent applications by enterprises obtained from the CNRDS database; the data on environmental regulation are from the China Environmental Statistical Yearbook and China Industrial Statistical Yearbook; the data on corporate social responsibility information disclosure come from the index scores of five stakeholders in the social responsibility report of listed companies in China by Hexun.com; and other control variables are from CSMAR database. We perform regression analysis of the balanced panel data after filtering with Stata 15.0.

### 3.2. Variable Definition

#### 3.2.1. Explained Variable-Enterprise Green Innovation (GI)

Referring to Yu, Zhang and Bi [71], we use the natural logarithm of the sum of enterprise green patent applications and 1 as the green innovation metric variable. The main considerations are as follows: First, it covers the patent application and authorization data for all listed enterprises in China, which are more stable and reliable; second, green patent applications can not only measure the status of the green innovation activities of enterprises but also has a high technical threshold. At the same time, it requires enterprises to develop, promote and apply corresponding green technologies based on improving the performance of their own products, which can reflect greater green innovation capabilities [72,73].

#### 3.2.2. Explanatory Variable–Environmental Regulatory Intensity (ERI)

Considering that differences in pollutant emissions between different industries will affect the comparability of environmental regulation intensity, and referring to the calculation method of environmental regulation intensity by Wang et al. [74], first, we collect data on three indicators, the ratios of the main pollutant emissions in industrial waste gas, waste water and solid waste to the total industrial output value and calculate the pollution emission per unit output of pollutants for each industrial industry. Second, we use Z-score standardization to process environmental regulation indicators and then standardize the unit pollution emissions of various industrial sectors and perform weighted average processing. Finally, we use the entropy method to construct a comprehensive evaluation and measurement system for environmental regulation. We believe that this calculation is reasonable as follows: First, the emission intensity of pollutants is an objective reflection of the intensity of environmental regulation. It is generally believed that the more serious the emission of pollutants in a certain industry, the stricter the corresponding environmental regulations will be; second, the three indicators included all forms of pollutants emitted by the industry; third, we use entropy weight to calculate the adjustment weight and comprehensively consider the differences in the proportions of different pollutant emissions caused by the differences in industrial structure between different industries.

#### 3.2.3. Moderator Variable-Enterprise Social Responsibility Information (CSR)

In order to avoid selectively disclosing social responsibility performance by whitewashing information, referring to the research of Zhang [75], we use the index results in the social responsibility reports of listed enterprises in China on Hexun.com to measure. We use entropy weight to adjust the weights of the five stakeholder responsibility indices published in the report: shareholders, employees, suppliers, customers and consumer rights organizations, the environment and society. Then we construct a comprehensive indicator system for social responsibility information disclosure after weighted averaging according to the above calculation method.

#### 3.2.4. Control Variable

In order to avoid the bias caused by omitted variables as much as possible, referring to Yu, Zhang and Bi [71] and Zhang [75], we select variables that are highly related to green innovation from multiple levels of the enterprise as control variables. In terms of enterprise financial characteristics: first, we select the size of the enterprise (Ln_Size) and use the natural logarithm of the enterprise’s total assets at the end of the year to measure. It is generally believed that the larger the enterprise scale, the higher the green innovation capability. Second, we select the enterprise profitability (ROA), which is measured by the return on total assets of the enterprise. The specific calculation method is the net profit divided by the average total assets. It is generally believed that a moderate debt operation is conducive to the long-term development of an enterprise and that it is convenient for the enterprise to have sufficient funds to improve technical equipment, improve production technology and carry out green innovation. Third, we select the growth capability of the company (Growth) and measure the growth rate of main business revenue. It is generally believed that the more opportunities for future growth of an enterprise, the better the development prospects and the greater the impetus for the enterprise to carry out technological innovation. Fourth, we select the enterprise maturity (Ln_Age) and use the natural logarithm of the sum of the enterprise’s listing years and 1 to measure. It is generally believed that the longer an enterprise has been established, the stronger its innovation. In terms of enterprise equity characteristics: First, we select the enterprise’s shareholding concentration (Top1) to measure as the shareholding ratio of the largest shareholder. The specific calculation method is the number of shares held by the largest shareholder divided by the number of outstanding shares of the enterprise. Second, we select the property rights of enterprises (State), in which the value of state-owned enterprises is 1, and otherwise the value is 0. In terms of the characteristics of enterprise boards: First, we select the size of the enterprise board of directors (Ln_Board) and use the logarithm of the total number of enterprise boards to measure. Second, we select the proportion of independent directors of the enterprise (Ind), that is, the ratio of the number of independent directors to the total number of directors. Third, we select the enterprise executive compensation (Ln_Pay) and use the natural logarithm of the sum of the annual salaries of the three highest paid executives of enterprises to measure.

All variable designs are shown in Table 1.

### 3.3. Model Design

Based on the above theoretical analysis and referring to Du et al. [76], we construct the following nonlinear fixed-effects regression models. The detailed reasons are as follows: (1) Fixed effect is the method that is suitable for comparing differences between variables and their interactions in research results. Essentially, it is also a method of controlling variables, which can prevent endogeneity problems associated with omitted variables and improves the accuracy of the estimated coefficients. (2) Time and industry fixed effects are designed to control for characteristics that are unique and do not vary with individuals at the time or industry levels. (3) Environmental regulation is a typical game process between government and enterprises, and the level of green innovation of enterprises varies with the intensity of environmental regulation. In order to test whether there is a nonlinear relationship between the intensity of environmental regulation and the green innovation of enterprises, this study introduces the quadratic term of the intensity of environmental regulation into the regression model:(1)$$GI_{i,t}=\beta_{0}+\beta_{1}ERI_{i,j,t}+\beta_{2}ERI^{2}{}_{i,j,t}+\beta_{3}X_{i,t}+\sum Year+\sum Industry+\varepsilon_{i,t}$$
(2)$$\begin{array}{l} GI_{i,t}=\beta_{0}+\beta_{1}ERI_{i,j,t}+\beta_{2}ERI^{2}{}_{i,j,t}+\beta_{3}CSR_{i,t}+\beta_{4}{\left(CSR*ERI\right)}_{i,j,t}+\beta_{5}{\left(CSR*ERI^{2}\right)}_{i,j,t}\\ +\beta_{6}X_{i,t}+\sum Year+\sum Industry+\varepsilon_{i,t} \end{array}$$

Among them, *i*, *j*, *t* respectively represent the enterprise, industry and time. The explained variable$\;GI_{i,t}$ represents the green innovation level of enterprise *i* in year *t*. The explanatory variable $ERI_{i,j,t}\;$represents the intensity of environmental regulation in year t of the industry j. In order to test the possible nonlinear relationship between environmental regulation and enterprise green innovation, we add the quadratic term of environmental regulation $\;ERI_{i,j,t}^{2}\;$. The moderator variable $CSR_{i,t}$ represents the social responsibility information disclosure of enterprises *i* in year *t*, ${\left(CSR*ERI\right)}_{i,j,t}\;$ represents the interaction item between enterprise social responsibility information disclosure and environmental regulation intensity. ${\left(CSR*ERI^{2}\right)}_{i,j,t}$ represents the interaction term between enterprise social responsibility information disclosure and the quadratic term of environmental regulation intensity. $X_{i,t}$ represents the other control variables, $\sum^{\;}Year$ and $\sum^{\;}Industry\;$ represent the year fixed effect and the industry fixed effect, respectively. $\varepsilon_{i,t}$ represents the random disturbance term.

### 3.4. Descriptive Statistics and Correlation Test

Table 2 reports the descriptive statistics of the main variables. As it can be seen, the mean of GI is 1.315, which is significantly larger than the median of 1.099. Its minimum is 0, and its maximum is 7.386, which indicates that the green innovation ability of most sample enterprises is lower than the average and that there is a significant gap in green innovation ability among sample enterprises. The average for environmental regulation is 0.173, and the median is 0.033, which indicates that the implementation of environmental regulation in most industries is lower than the average level. The difference between the minimum value and the maximum value indicates that the investment in environmental regulation varies greatly among industries. The average enterprise social responsibility information disclosure is 0.508, and the maximum is only 4.093, which indicates that the social responsibility information disclosure score of Chinese listed companies given by Hexun.com is generally low and the quality of Chinese listed companies’ social responsibility information disclosure needs to improve. In addition, the descriptive statistics of other control variables are all within a reasonable range. Table 3 reports the results of the correlation test between the main explanatory variables and the explained variables. As can be seen, except the correlation coefficient between enterprise scale and enterprise green innovation level, which is 0.54, the correlation coefficients between variables are less than 0.5, and the variance inflation factor (VIF) of each variable is less than 10, which shows that there is no serious multicollinearity problem among the variables and it has no effect on the regression model.

## 4. Regression Analysis

Column (1) of Table 4 reports the test results of the impact of environmental regulation intensity on enterprise green innovation. It shows that the coefficient between $ERI^{2}$ and GI is 0.097 and the coefficient between ERI and GI is −0.501, both of which are significant at the level of 1%, indicating that there is a U-shaped relationship between environmental regulation and enterprise green innovation. As shown in Figure 2, H1a is verified. Column (2) of Table 4 shows that the coefficient between GI and $ERI^{2}$ in the first lag period is 0.121 and the coefficient between ERI and GI in the first lag period is −0.455, both of which are significant at the level of 1%, indicating that the environmental regulation in the lag period has a more significant impact on enterprise green innovation. H1b is verified. However, it is not rigorous to conclude that there is a U-shaped relationship only if the coefficient of the square terms of the variables is statistically significant, so we further verify the U-shaped results. We omit the control variables in the benchmark regression model (1) and derive it to obtain Formula (4). We can obtain the extreme point of ERI, 2.600, which is within the range of (2.347,2.940) and satisfies the condition of the U-shaped curve.

Column (3) of Table 4 reports the regulatory effect of social responsibility information disclosure. The regression results show that the coefficient between CSR and $ERI^{2}\;$is 0.068, and the coefficient between CSR and ERI is −0.370, both of which are significant at 1%, indicating that social responsibility information disclosure significantly adjusts the relationship between environmental regulation and enterprise green innovation. As shown in Figure 3, H2 is verified.

Drawing on the analysis method of Tang et al. [77] of the regulatory effect, we make the following descriptions from three aspects: the shape of the U-shaped curve, the inflection point and the overall level. First, we analyze the shape of the U-shaped curve between environmental regulation and corporate green innovation, which depends on the curvature of the apex. We omit the control variable of Formula (2) and simplify it to Formula (5) and then perform a quadratic derivation of ERI to obtain the vertex curvature K, as shown in Formula (6). If the vertex curvature K is greater than 0, the larger K is, the steeper the U-shaped curve is and vice versa. The influence of the moderator variable on the shape of the curve depends on the sign of the coefficient of $\beta_{5}$ after the partial derivative of K with respect to the CSR. If $\beta_{5}$ is positive, the larger the CSR, the larger the K and the steeper the U-shaped curve and vice versa. In column (3) of Table 4, $\beta_{5}$ is 0.042, and it is significant at 1%, which shows that the high-quality social responsibility information disclosure behavior of enterprises is conducive to strengthening internal and external information communication, strengthening the U-curve relationship between environmental regulation and enterprise green innovation and making it steeper.
(3)$$GI=\beta_{0}+\beta_{1}ERI+\beta_{2}ERI^{2}$$
(4)$$GI^{\prime }=\beta_{1}+2\beta_{2}ERI$$

Secondly, we analyze the influence of the regulatory variable CSR on the inflection point of the U-shaped curve. We take the first derivative of ERI in Formula (5) and set it to 0, so that the value Formula (7) of the curve inflection point ERI* is obtained. We further analyze the impact of CSR changes on ERI* by taking the partial derivative of Formula (7). If the partial derivative is less than 0, the larger the adjustment variable CSR is, the smaller the ERI* is. At this point, the curve inflection point moves to the left and vice versa. The regression results in column (3) show that $\;\beta_{1},\;\;\beta_{2},\;\;\beta_{4},\beta_{5}\;$ are all significant, and $\left(\beta_{1}\beta_{5}-\beta_{2}\beta_{4}\right)$ after partial derivatives is −0.0043, which is less than 0, which means that the greater the CSR, the more the curve inflection point will move to the left. In other words, high-quality enterprise social responsibility disclosure can send favorable signals to stakeholders, form a supervisory mechanism internally and externally, improve the innovative compensation effect brought by environmental regulation and shift the U-shaped inflection point to the left.
(5)$$\begin{array}{ll} GI & =\beta_{0}+\beta_{1}ERI+\beta_{2}ERI^{2}+\beta_{3}CSR+\beta_{4}(CSR*ERI)+\beta_{5}(CSR*ERI^{2})\\ & =\beta_{0}+\beta_{1}ERI+(\beta_{2}+\beta_{5}CSR)*ERI^{2}+\beta_{3}CSR+\beta_{4}(CSR*ERI) \end{array}$$
(6)$$K=GI^{\prime \prime }=2(\beta_{2}+\beta_{5}CSR)$$
(7)$$ERI^{*}=-(\beta_{1}+\beta_{4}CSR)/2*(\beta_{2}+\beta_{5}CSR)$$

Finally, we analyze the influence of the moderator variable (social responsibility information disclosure) on the overall U-shaped curve. We record high-quality social responsibility information disclosure as $GI_{CSRH}$ and low-quality social responsibility information disclosure as $GI_{CSRL}$. If $GI_{CSRH}-GI_{CSRL}$ is constantly greater than 0, it means that high-quality social responsibility disclosure has improved the overall level of the relationship between environmental regulation and the enterprise green innovation. In order to make the function $\beta_{4}ERI+\beta_{5}ERI^{2}+\beta_{3}$ in Formula (8) be constantly greater than 0 when ERI takes any value, it needs to be satisfied that ① $\;\beta_{5}>0$; ② the function has no real root, that is $\beta_{4}{}^{2}-4\beta_{3}\beta_{5}<0$. In the regression result of Formula (8), $\beta_{5}$ was 0.1899, which is significant at the level of 1% and satisfies the condition ①. Both $\beta_{3}\;\mathrm{and}\;\beta_{4}$ are significant at 1% and $\beta_{4}{}^{2}-4\beta_{3}\beta_{5}=-0.002<0$, which satisfies the condition ②. Therefore, when ERI takes arbitrary values, $GI_{CSRH}-GI_{CSRL}$ is constantly greater than 0. H2 is further verified. In other words, in a period when environmental regulation is weak and its intensity is rising, high-quality social responsibility disclosure enables stakeholders to better understand the implementation of environmental regulation by enterprises through perception and strengthen their legitimacy identification. Additionally, the beneficial signals released by enterprises to the capital market can help them form a certain reputation capital. When the intensity of environmental regulation is high, the compensation effect of green innovation comes into play. At this time, high-quality social responsibility disclosure will further strengthen the relationship between enterprises and stakeholders and provide long-term financial support for enterprises on the basis of environment-friendly innovation.
(8)$$GI_{CSRH}-GI_{CSRL}=(\beta_{4}ERI+\beta_{5}ERI^{2}+\beta_{3})(CSR_{H}-CSR_{L})$$

Column (4) and column (5) of Table 4 respectively report the impacts of non-state-owned enterprises and state-owned enterprises on enterprise green innovation under different intensity of environmental regulation. The results show that in non-state-owned enterprises, the regression coefficient between $ERI^{2}$ and GI is 0.04, the regression coefficient between ERI and GI is −0.479, both of which pass the significance test at 1%. In state-owned enterprises, the regression coefficient between $ERI^{2}$ and GI is 0.069, the regression coefficient between ERI and GI is −0.921, which are also significant at 1%. It shows that with the strengthening of government environmental regulation, the overall level of green innovation of non-state-owned enterprises is higher than that of state-owned enterprises and the inflection point comes earlier. As shown in Figure 4, H3 is verified. It can be seen that non-state-owned enterprises without innate advantages in the relationship between government and enterprises attach great importance to a good relationship with the government, show compliance with the environmental regulations within the acceptable range and expect to obtain economic returns from the government, such as taxation relief, financial subsidies, etc. There is a natural relationship between state-owned enterprises and the government. State-owned enterprises have made great contributions to local GDP, taxation, employment and public services, which make state-owned enterprises less likely to respond when environmental pollution is not serious and the intensity of environmental regulation is low. However, when the environmental problems become more severe and the intensity of environmental regulation increases, in order to alleviate the pressure of public opinion, state-owned enterprises show a positive attitude towards environmental protection and a green responsibility image to the public, thereby increasing the output of green innovation.

Column (6) of Table 4 reports the regression results when CSR is added as the moderator variable in non-state-owned enterprises. In this part, we empirically test the influence of the quality of social responsibility information disclosure on the relationship between environmental regulation and enterprise green innovation in non-state-owned enterprises. The regression results show that the coefficient between CSR and $ERI^{2}$ is 0.049, and the coefficient between CSR and ERI is −0.233, both of which are significant at 5%. Compared with general non-state-owned enterprises, non-state-owned enterprises that disclose high-quality social responsibility information have a stronger regulatory effect on the relationship between environmental regulation and green innovation, which can carry out green innovation activities earlier and put green production processes into production and use. H3 is verified. It can be seen that the institutional pressure on non-state-owned enterprises that disclose social responsibility information is significantly greater than that on general non-state-owned enterprises, which leads to the earlier appearance of the inflection point of the U-shaped curve, as shown in Figure 5. 

Combining the above empirical results, Table 5 reports the validation relationships and empirical results for all hypotheses.

## 5. Robustness Test

### 5.1. Replacement Variable Method

In this paper, we replace the ratio of corporate green patent applications to total patent applications with the natural logarithm of the sum of corporate green patent applications with 1 as a measurement variable for green innovation. Then we perform a regression test on the full sample. The results show that the regression coefficient between $ERI^{2}$ and GI is 0.057 and the regression coefficient between ERI and GI is −0.738, both of which are significant at 1%, as shown in column (1) of Table 4. The regression coefficient between CSR and $ERI^{2}\;$is 0.097, which is significant at 5%, as shown in columns (1) and (2) of Table 5. It is basically consistent with the above research conclusions.

### 5.2. Endurance Test

Further, we use the method of propensity score matching (PSM) to test the endogeneity between environmental regulation and enterprise green innovation. We define a dummy variable between 0 and 1 with the median of the green innovation level and use Logit regression to match the listed companies in each sample. We chose the green innovation level as the matching variable. Since the descriptive statistics show that the median of the green innovation level is 3.555, we divide the observations with the green innovation level less than or equal to 3.555 and the observations greater than 3.555 into two groups for matching. The result shows that the regression coefficient between $ERI^{2}$ and GI is 0.039 and the regression coefficient between ERI and GI is −2.276, both of which are significant at 1%, as shown in column (3) of Table 6. This is basically consistent with the above research conclusions.

## 6. Discussion

In the current and lag effects, the impact of environmental regulation intensity (ERI) on enterprise green innovation shows a U-shaped nonlinear characteristic of first inhibiting and then promoting. That is, when the intensity of ERI is low, stronger ERI by the government will cause the cost of pollution control to rise for enterprises, leading to reduced profits and squeezing out R&D funds, thus inhibiting enterprise green innovation. When the intensity of ERI is high, the pressure of excessive pollution abatement costs forces enterprises to increase green technology investment to form competitive advantage, thus promoting green innovation. The above conclusions are relatively close to the research conclusions of some scholars and slightly different depending on the sample selection, the study period and the model design. Scholars’ research conclusions on the relationship between environmental regulation and green innovation focus on the U, inverted U nonlinear and linear relationships. For example, Li et al. [27] collected time-series data by the Chinese government (2000–2018) and analyzed the mechanism of action of the factors influencing the GTIB of construction enterprises, and they found that the role of environmental regulation in the GTIB of construction enterprises is nonlinear. Renqiao et al. [78] selected 30 provinces’ industrial enterprises in 2008–2018, built a static panel and dynamic GMM panel model, and analyzed the impacts of heterogeneous environmental regulations on green innovation, showing that the environmental regulation has a U-shaped relationship with the efficiency of green technology research and development, which has the incentive effect of first inhibiting and then promoting, but command environmental regulation has inverted U and U shape relationship respectively on green science and technology development and transformation efficiency. Mingyue et al. [28] used the data of 2278 manufacturing enterprises in China, divided green technological innovation into green process innovation and green product innovation and verified the nonlinear relationship between environmental regulation and enterprise green technological innovation, showing there is an inverted U-shaped relationship between them. This conclusion will not change due to the types of green technological innovation, while the impact of environmental regulation on enterprise green product innovation is greater than that of green process innovation. Zhaoqiang et al. [18] used the sample data of A-share listed companies from 2010 to 2019, took the promulgation of the new environmental protection law (NEPL) in 2014 as the exogenous shock of a quasi-natural experiment and tested the relationship of environmental regulation with enterprise green innovation, finding the NEPL has significantly promoted the green innovation of heavily polluting enterprises, and the marginal effects of NEPL exhibit a fluctuating trend of first decline, then rise and then decline over time, and the overall trend is downward. Berrone et al. [8] used environment-related patent data of 326 publicly traded firms from polluting industries in the United States and verified that institutional environment pressures can trigger such innovation, especially in those firms displaying a greater deficiency gap. Wu Bao et al. [21] used a sample of 4924 private Chinese companies indicating that both formal and informal regulation pressures have a positive effect on green innovation. Zhao Li et al. [41] selected Chinese provincial data from 2009 to 2018 and explored the influence of different public participation constraints on green technological innovation, showing that as the constraints of news media increase by 1 unit, the green technological innovation will increase by 0.210 units. However, the ENGO constraints have not effectively promoted green technological innovation. Different from athe bove studies, this paper uses A-share listed companies from 2010 to 2020, confirms the U-shaped relationship between current and lagging environmental regulation and green innovation and finds that environmental regulation has a greater impact on the green innovation of non-state-owned enterprises, supporting and enriching Porter’s hypothesis from the micro level.

In terms of the regulatory effect, this study shows that social responsibility information disclosure has a significant regulatory effect on environmental regulation intensity and enterprise green innovation, and non-state-owned enterprises have a stronger regulating effect, which is obviously different from the conclusions of the existing literature. For example, Kraus et al. [16] used data from 297 large manufacturing firms in Malaysia and investigated the influence of corporate social responsibility (CSR) on environmental performance, showing that CSR has no direct significant influence on environmental performance but is positively correlated with environmental strategy and green innovation, which again improves environmental performance. Zhaoqiang et al. [18] showed that environmental responsibility plays a mediating role in the relationship between environmental regulation and enterprise green innovation. Zhao Li et al. [41] found that environmental regulatory enforcement has not yet played a positive moderating role in the relationship between news media constraints, community resident constraints and green technological innovation, but only played a positive moderating role in the relationship between ENGOs constraints and green technological innovation. Doshi et al. [44] examined how organizational characteristics moderate establishments’ responses to a prominent environmental information disclosure program, finding particularly rapid improvement among establishments located close to their headquarters and among establishments with proximate siblings. Peter et al. [61] used a sample of 191 firms from the five most polluting industries in the US, verifying a positive association between environmental performance and the level of discretionary environmental disclosures. Zhaoqiang et al. [18] found that the NEPL plays a more significant role in promoting the green innovation of state-owned enterprises. Wu Bao et al. [21] showed that political connections positively moderate the effect of formal regulation pressure on green innovation but negatively moderate the effect of informal regulation pressure. By comparison, the conclusions of this paper not only confirm the interaction of environmental regulation and social responsibility information disclosure but also find influencing differences of enterprises with heterogeneous property rights, which provides rich evidence for exploring the interactive relationship between environmental regulation, social responsibility information disclosure and green innovation.

## 7. Conclusions 

In this paper, we selected Chinese A-share listed companies on the Shanghai and Shenzhen Stock Exchanges from 2010 to 2020 as our samples and discussed the driving mechanism of government environmental policy pressure on enterprise green innovation and the regulatory effect of CSR disclosures. The research concludes that the impact of environmental regulation on green innovation in enterprises presents U-shaped characteristics and that the impact of environmental regulation with a lag period is more significant. Among them, non-state-owned enterprises are more sensitive to increasing intensity of environmental regulation. The quality differences between CSR disclosures have an impact on the relationship between environmental regulation and enterprise green innovation, and non-state-owned enterprises that disclose high-quality CSR information can improve the overall U-shaped curve of environmental regulation and enterprise green innovation. The conclusions obtained can provide certain theoretical guidance and practical basis for governments to optimize and adjust the intensity of environmental regulation, improve environmental regulation policies and scientifically formulate governance measures for enterprises to improve the level of social responsibility information disclosure and achieve ecological protection and high-quality development.

In terms of the state and the government: Firstly, the state and the central government must always adhere to the concept that green innovation leads to high-quality development and formulate appropriate environmental regulations to effectively stimulate the development of green innovation activities by enterprises. Because different environmental regulations have different effects on the green innovation of enterprises. Therefore, the government cannot simply use a one-size-fits-all model when implementing environmental regulation policies. It should timely adjust and improve enterprises with different attributes and industries and adopt differentiated environmental regulation strategies. For example, the government can mobilize enthusiasm for green innovation by appropriately exerting legal pressure on state-owned enterprises and increasing the assessment of their green R&D achievements. Secondly, local governments should strengthen the supervision of environmental governance to ensure the effective implementation of the central government’s environmental protection policies. In particular, in the face of environmental regulations crowding out green innovation funds for enterprises, the government should flexibly develop channels to encourage enterprise green R&D. For example, for enterprises with short-sighted management, the government should give them greater tax incentives and financial subsidies and actively guide enterprises to transform into green innovation to achieve a win–win situation between environmental governance and green development. Thirdly, the state should constantly regulate the social responsibility reporting requirements of listed companies and refine and improve the disclosure standards to better supervise and restrain the opportunistic behavior of management and ensure that enterprises provide true and reliable social responsibility information.

In terms of enterprises: Firstly, enterprises should actively cooperate with and implement the government’s environmental regulations, fully recognize the importance of green innovation to the compensation effect of government environmental regulations and incorporate green innovation into their long-term strategic plans. Secondly, enterprises should formulate a complete mechanism of talent introduction, improve the construction of supporting facilities for their research and development and make adequate preparations for the development of their own green innovation and the improvement of the level of green innovation. Thirdly, enterprises should internalize social responsibility commitment as the core element of their own development, actively maintain relationships with internal and external stakeholders, improve the quality of social responsibility information disclosure and prevent management from disclosing social responsibility information out of self-interest motives. In particular, non-state-owned enterprises can improve their nonfinancial performance by improving the quality of social responsibility disclosure to improve the market competitiveness of enterprises.

As with most empirical research, our study had three limitations that provide potential directions for future research. First, in order to measure the intensity of environmental regulation, our study constructed a comprehensive evaluation and measurement system with the help of Z-score standardization and entropy, which can weaken the influence of measurement error to a certain extent. However, with the improvement and development of environmental regulation policies, its measurement method is multidimensional and dynamic. For example, command and control, market-incentive and public-participation environmental regulation can be classified according to the effect and degree of influence on corporate behavior and decision making. Therefore, in the environment of the extensive empowerment of social responsibility information disclosure, how the incentive mechanism for the green innovation of different types of environmental regulations needs to be further studied. Second, considering that the performance of corporate green innovation is closely related to factors such as digital strategic orientation and executive environmental awareness, how to study its possible moderating effects from the perspective of digital investment and executive characteristics needs to be further tested. Third, our study examines the differential impact of environmental regulation on green innovation under the heterogeneity of enterprise property rights. Considering the correlation between the growth stage and marketization degree of enterprises and the performance of green innovation, the research on how to identify the differentiated impact of environmental regulation and enterprise green innovation based on different growth stages and marketization degree needs to be enriched.

## Figures and Tables

**Figure 1 ijerph-19-14771-f001:**
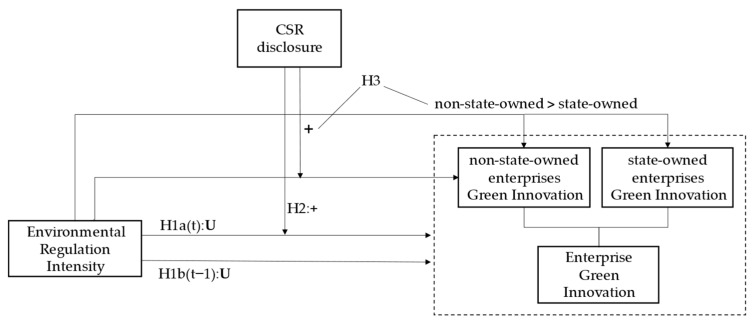
Theoretical model.

**Figure 2 ijerph-19-14771-f002:**
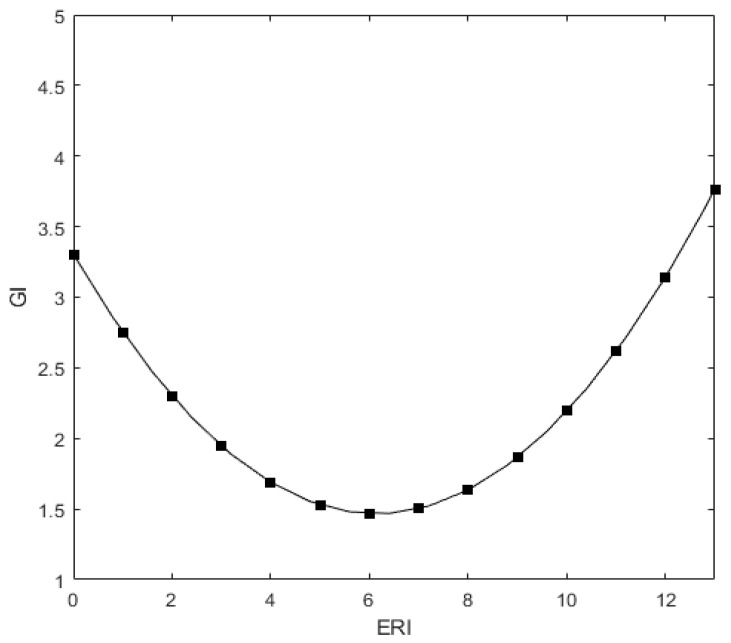
U-shaped relationship diagram between environmental regulation and enterprise green innovation.

**Figure 3 ijerph-19-14771-f003:**
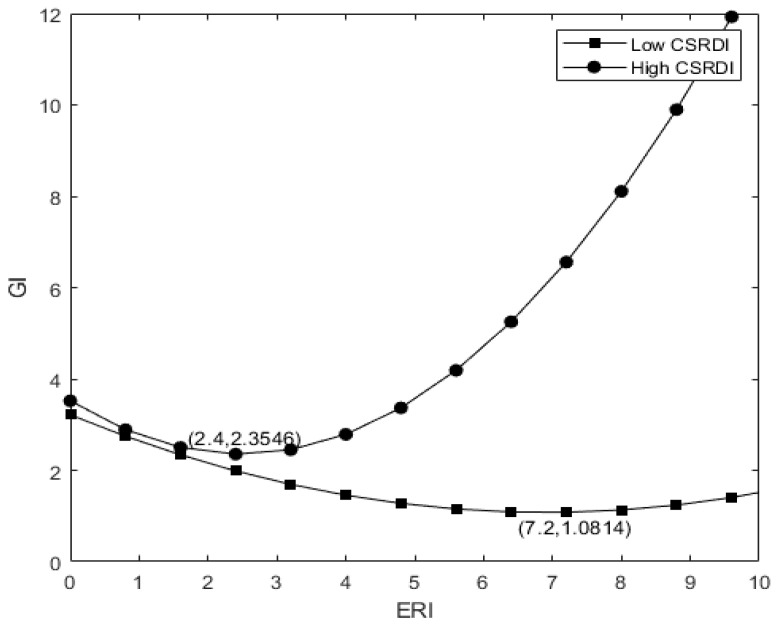
Regulatory Effect Diagram of Social Responsibility Information Disclosure.

**Figure 4 ijerph-19-14771-f004:**
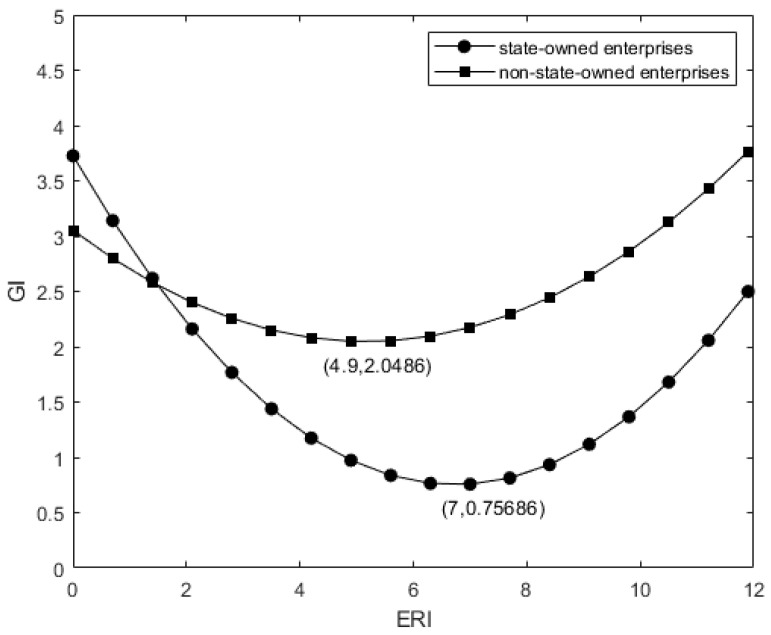
Comparison of the U-shaped relationship between environmental regulation and green innovation in state-owned and non-state-owned enterprises.

**Figure 5 ijerph-19-14771-f005:**
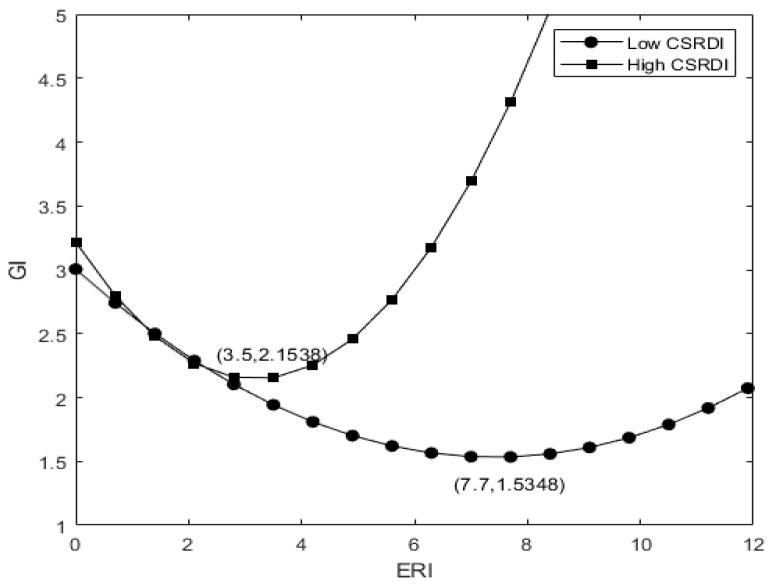
The regulatory effect diagram of non-state-owned enterprises’ social responsibility information disclosure.

**Table 1 ijerph-19-14771-t001:** Variable Definition and Description.

Type of Variable	Variable Name	Variable Symbol	Variable-Definition
Explained variable	Enterprise green innovation	GI	The natural logarithm of green patent applications plus 1
explanatory variable	Environmental regulation intensity	ERI	The ratio of the exhaust gas emission to the total industrial output
			The ratio of the wastewater discharge to the total industrial output
			The ratio of the solid waste pollution to the total industrial output
Moderator variable	Social responsibility information	CSR	Shareholder liability index
			Employee responsibility index
			The supplier, customer, and consumer equity liability index
			Environmental responsibility index
			Social responsibility index evaluation
Control variable	Scale	Ln_Size	The total assets of the enterprise at the end of the year take the natural log
	Profitability	ROA	Net profit is divided by the average total assets
	Debt capacity	Lev	Total liabilities at the end of the year are divided by the total assets at the end of the year
	Growth ability	Growth	Main business revenue growth rate
	Enterprise maturity	Ln_Age	Natural logarithm of listing years plus 1
	Equity concentration	Top1	The ratio of the largest shareholder
	Board size	Ln_Board	The total number of enterprise board of directors is the natural log
	Proportion of enterprise independent directors	Ind	The proportion of the number of independent directors to the total number of the board of directors
	Enterprise executive compensation	Ln_Pay	The sum of the three most paid executives takes the natural logarithm
	Property nature	State	The value of state-owned enterprises is 1, otherwise the value is 0

**Table 2 ijerph-19-14771-t002:** Descriptive statistics.

Variable	N	Mean	Median	Min	Max	SD
GI	7205	1.315	1.099	0.000	7.386	1.328
ERI	7205	0.173	0.033	0.000	5.891	0.393
CSR	7205	0.508	0.106	0.023	4.093	0.957
Ln_Pay	7205	14.345	14.323	11.132	17.746	0.709
Ln_Board	7205	2.162	2.197	1.386	2.833	0.188
Growth	7205	0.222	0.155	−0.467	17.773	0.573
Ln_Size	7205	22.351	22.190	19.541	28.636	1.240
Top1	7205	0.343	0.320	0.034	0.900	0.146
Ln_Age	7205	2.314	2.485	0.000	3.689	0.705
ROA	7205	0.042	0.037	−1.125	0.478	0.066
Ind	7205	0.370	0.333	0.200	0.800	0.055
Lev	7205	0.432	0.432	0.008	2.155	0.199

**Table 3 ijerph-19-14771-t003:** Correlation analyses.

	GI	ERI	CSR	Ln_Pay	Ln_Board	Growth	Ln_Size	Top1	Ln_Age	ROA	Ind	Lev
GI	1											
ERI	−0.14 ***	1										
CSR	0.05 ***	0.11 ***	1									
Ln_Pay	0.33 ***	−0.13 ***	0.06 ***	1								
Ln_Board	0.13 ***	0.06 ***	0.13 ***	0.08 ***	1							
Growth	0.03 **	−0.009	0.05 ***	0.04 ***	0.03 ***	1						
Ln_Size	0.54 ***	0.02 *	0.16 ***	0.48 ***	0.27 ***	0.05 ***	1					
Top1	0.08 ***	0.10 ***	0.12 ***	0.03 ***	0.05 ***	0.10 ***	0.25 ***	1				
Ln_Age	0.17 ***	−0.13 ***	−0.05 ***	0.23 ***	0.04 ***	−0.07 **	0.31 ***	−0.06 **	1			
ROA	−0.03 ***	0.06 ***	0.11 ***	0.19 ***	0.07 ***	0.06 ***	0.02 *	0.16 ***	−0.14 ***	1		
Ind	0.03 ***	−0.003	0.02 *	0.04 ***	−0.42 ***	0.01	0.06 ***	0.07 ***	0.04 ***	−0.04 **	1	
Lev	0.27 ***	0.03 ***	0.07 ***	0.06 ***	0.14 ***	0.05 ***	0.45 ***	0.04 ***	0.27 ***	−0.38 **	0.02	1

Note: * *p* < 0.1, ** *p* < 0.05, and *** *p* < 0.001.

**Table 4 ijerph-19-14771-t004:** Regression results of the main test.

	(1)	(2)	(3)	(4)	(5)	(6)
	GI	GI	GI	GI	GI	GI
ERI^2^	0.097 *** (5.26)		0.068 *** (3.26)	0.040 *** (4.83)	0.069 *** (10.12)	0.028 ** (2.97)
ERI	−0.501 *** (−6.99)		−0.370 *** (−4.47)	−0.479 *** (−7.25)	−0.921 *** (−13.21)	−0.402 *** (−5.40)
CSR			0.016 * (0.89)			0.057 ** (2.28)
CSR × ERI			−0.168 *** (−3.16)			−0.233 ** (−2.83)
CSR × ERI^2^			0.042 *** (2.61)			0.049 ** (2.49)
L.ERI^2^		0.121 *** (4.15)				
L.ERI		−0.455 *** (−6.58)				
Ln_Pay	0.093 *** (4.27)	0.096 *** (4.12)	0.097 *** (4.42)	0.275 *** (9.45)	0.120 *** (3.62)	0.277 *** (9.47)
Ln_Board	0.157 ** (2.02)	0.139 (1.67)	0.167 ** (2.15)	0.277 ** (2.41)	−0.121 (−1.11)	0.264 ** (2.29)
Growth	−0.027 (−1.23)	−0.027 (−1.13)	−0.028 (−1.27)	−0.006 (−0.63)	−0.079 ** (−2.23)	−0.005 (−0.59)
Ln_Size	0.570 *** (39.23)	0.583 *** (37.97)	0.571 *** (38.80)	0.580 *** (24.70)	0.696 *** (36.10)	0.575 *** (24.19)
Top1	−0.167 * (−1.84)	−0.174 (−1.79)	−0.165 * (−1.81)	0.074 (0.61)	−0.437 ** (−3.17)	0.074 (0.61)
Ln_Age	−0.020 (−0.90)	−0.032 ** (−1.21)	−0.014 (−0.66)	−0.054 (−1.81)	0.049 (1.37)	−0.053 * (−1.76)
ROA	0.198 (0.89)	0.226 *** (0.98)	0.192 (0.87)	1.601 *** (5.57)	0.893 ** (2.59)	1.565 *** (5.44)
Ind	−0.114 (0.18)	−0.021 (−0.08)	0.073 (0.29)	0.940 * (2.43)	−1.177 *** (−3.41)	0.882 ** (2.27)
Lev	0.286 *** (3.58)	0.336 *** (3.97)	0.274 *** (3.44)	0.508 *** (4.50)	−0.104 (−0.91)	0.503 *** (4.46)
_cons	−12.858 *** (−13.535)	−13.081 *** (−35.27)	−12.987 *** (−36.50)	−14.216 *** (−25.57)	−12.373 *** (−25.72)	−14.091 *** (−24.89)
Industry	control	control	control	control	control	control
Year	control	control	control	control	control	control
Adj R^2^	0.389	0.465	0.390	0.406	0.529	0.406
N	7205	6584	7205	4033	3172	4033

Note: * *p* < 0.1, ** *p* < 0.05, and *** *p* < 0.001.

**Table 5 ijerph-19-14771-t005:** Results of the hypothesis test.

Hypothesis	Significance	Impact
H1a: There is a U-shaped curve relationship between the intensity of environmental regulation and enterprises’ green innovation.	Significant	U
H1b: Environmental regulation with a lag period has a more significant impact on green innovation.	Significant	U
H2: CSR disclosure has a positive regulatory effect on the relationship be-tween environmental regulation and enterprise green innovation.	Significant	positive
H3: Environmental regulation and CSR disclosure has a greater impact on the green innovation of non-state-owned enterprises.	Significant	positive

**Table 6 ijerph-19-14771-t006:** Results of the robustness test.

	(1)	(2)	(3)
	GI	GI	GI
ERI^2^	0.057 ***	0.030 ***	0.167 ***
	(7.40)	(3.33)	(8.17)
ERI	−0.738 ***	−0.499 ***	−2.276 ***
	(−10.49)	(−6.02)	(−9.33)
CSR		0.158 ***	
		(6.14)	
CSR × ERI		−0.459 ***	
		(−6.23)	
CSR × ERI^2^		0.097 **	
		(4.41)	
Pay	0.217 ***	0.210 ***	0.364 ***
	(6.72)	(6.47)	(4.35)
Board	0.305 **	0.301 **	0.484 *
	(2.67)	(2.64)	(1.79)
Growth	−0.082 ***	−0.081 ***	−0.056
	(−6.35)	(−6.29)	(−0.72)
Size	0.346 ***	0.332 ***	0.849 ***
	(16.13)	(15.32)	(15.03)
Top1	0.286 **	0.292 **	−0.339
	(2.13)	(2.19)	(−1.01)
Ln_Age	−0.154 ***	−0.151 ***	−0.025 **
	(−4.77)	(−4.65)	(−0.28)
ROA	1.835 ***	1.786 ***	1.515 *
	(3.96)	(5.47)	(1.72)
Ind	0.150	0.095	0.264
	(2.02)	(0.25)	(0.31)
Lev	0.098	0.076	0.100
	(0.83)	(0.65)	(0.36)
_cons	−8.548 ***	−8.216 ***	−24.692 ***
	(−16.77)	(−15.67)	(−18.17)
Industry	control	control	control
Year	control	control	control
Adj R^2^	0.1544	0.1599	0.2845

Note: * *p* < 0.1, ** *p* < 0.05, and *** *p* < 0.001.

## Data Availability

The data of this research are publicly available.

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
