# Peer review of "Environmental Regulation, Corporate Social Responsibility (CSR) Disclosure and Enterprise Green Innovation: Evidence from Listed Companies in China"

_ijerph, 2022, doi:10.3390/ijerph192214771_

Round 1

Reviewer 1 Report

The corporate social responsibility and the green innovation remains an emerging topic.

Writing needs to be more careful, (E.g. line 16, line 309 – ST?)

There are sentences without references.

There are sentences that are too big and that make them confusing. (Suggestion: shorter and more direct sentences).

The results could be present clearer (e.g. “Column (6)” but it’s not mention the table).

In the European Union we have a definition of micro-enterprises, probably in China it is different. This sentence: “This paper conducts an in-depth investigation from the perspective of micro-enterprises and finds that…”.  But in the sample: “Finally, we obtained 11-year continuous data of 655 listed companies, a total of 7205 research samples. Among them, the industry division refers to the 2012 edition of the SEC industry classification standards, involving three major industries, including mining, manufacturing, electricity, heat, gas and water production and supply.”

Author Response

Dear editors and reviewers:

A Report of Amendments and Responses to the Editor’s and Reviewers’ Comments

We are grateful to you for allowing us to revise and resubmit the paper. We would also like to thank the anonymous reviewer for his/her insightful and constructive comments. We have fully revised the paper in line with all your recommendations, which have helped us in further improving the quality of the current version of the paper. We have outlined below the specific amendments that we have carried out in response to each of your suggestions. We hope that we have succeeded in addressing the concerns raised in your comments.

Below, we detail our point-by-point responses to your comments (in bold and italic fonts). Our response starts with ***. All revisions are highlighted in the manuscript.

Reviewer-1 Comments:

Comments to the Author

Comments and Suggestions for Authors

1.The corporate social responsibility and the green innovation remains an emerging topic. Writing needs to be more careful, (E.g. line 16, line 309 – ST?)

*** We are very grateful to you for these positive comments on the academic and practical relevance of our paper. First, we revise the writing (line-16-20): “Based on Chinese A-share listed companies in Shanghai and Shenzhen Stock Exchange from 2010 to 2020, this study explores the influencing mechanism of environmental regulation on corporate green innovation and the moderating mechanism of CSR disclosure by constructing a nonlinear fixed-effect regression model.”

In addition, we have added footnotes to "ST" and "*ST" in line 309-310 (page 7): “ST is the abbreviation of special treatment, which refers to the stock of a domestic listed company that has suffered losses in operation for two consecutive years and has been warned of the risk of delisting.” “*ST” refers to a stock that is about to be delisted, when a listed company has been operating for three consecutive years of losses, or less than three years but has serious financial losses, then the stock name will be preceded by "*" in addition to "ST".”

  1. There are sentences without references.

*** We are thankful for this constructive comment and have added references to the following sentences:

“As a regulatory pressure, environmental regulation transforms environmental protection needs into strict policies by providing normative content and is considered an important measure to achieve green development of enterprises [4].” (line 55)

“Internal governance mainly emphasizes the corporate management's values [5] and social responsibility cognition [6].” (line 56)

“Faced with the same external institutional pressure, enterprises have different degrees of green innovation, which to a certain extent depends on the attitude of actively seeking social responsibility within the enterprise [7].” (line 59)

“Studies have shown that enterprises that can take into account both social and environmental issues are more able to persist in developing green innovation in the long run [29].” (line 125)

  1. There are sentences that are too big and that make them confusing. (Suggestion: shorter and more direct sentences).

*** We apologize for the language problems in the original manuscript. The language presentation was improved with shorter and more direct sentences as the following:

For example:“In summary, although there are valuable results in the literature on environmental regulation and green innovation, no consensus has been reached in the academic community so far. In order to achieve green innovation incentives for enterprises, it has become a challenge for the government to optimally adjust and choose the appropriate level of environmental regulation enforcement according to their actual situation and differences.” (line116-121)

“Berchicci et al. [36] pointed out that if an enterprise actively fulfills its social responsibilities such as environmental protection, it will have more opportunities to gain social attention and receive environmental subsidies from the relevant government departments, thus greatly increasing the intensity of the enterprise's environmental investment and promoting the level of green technology and product updates.” (line142-146)

  1. The results could be present clearer (e.g. “Column (6)” but it’s not mention the table).

*** We are extremely grateful for the comment. In order to present the results clearer, we have detailed the table in which each column is located:

“Column (2) of Table 4” (line 459)

“Column (3) of Table 4” (line 471)

“Column (4) and column (5) of Table 4” (line 535)

“Column (6) of Table 4” (line 562)

  1. In the European Union we have a definition of micro-enterprises, probably in China it is different. This sentence: “This paper conducts an in-depth investigation from the perspective of micro-enterprises and finds that…”.  But in the sample: “Finally, we obtained 11-year continuous data of 655 listed companies, a total of 7205 research samples. Among them, the industry division refers to the 2012 edition of the SEC industry classification standards, involving three major industries, including mining, manufacturing, electricity, heat, gas and water production and supply.”

*** We regret that the inaccuracies in the English presentation of the original manuscript have caused experts to misunderstand “micro-enterprises”. We explain it in detail below. “Based on different perspectives, scholars have explored the influencing factors of green innovation. Some of them focus on the macro level, such as “industries” and “regions”, while others focus on the micro level, such as “enterprises”. The term "micro-enterprises" is defined in original manuscript from this perspective, rather than based on the size of enterprises. To avoid misunderstanding, we have replaced the term "micro-enterprises" with "enterprises".” (line 629-630)

Reviewer 2 Report

The subject of the article is to analyze the impact of environmental regulations and CSR disclosure on ecological innovations in enterprises.

Analysis based on data from 655 companies listed on stock exchanges in China in 2010-2020.

The linear regression method was used in the data analysis.

The subject of the article is important due to the need to protect the environment.

Remarks:

No purpose of the article, no research method in the abstract.

The ESG acronym mentioned in the keywords is not developed and nowhere else is used.

In the article, there is no justification for the selection of a sample of companies, no justification for the selection of the data analysis method (linear regression), no justification for the selection of variables.

What does it mean that the hypothesis has been verified? Positive or negative?

The H4 hypothesis is not formulated and is being verified.

Author Response

Dear editorsand reviewers:

A Report of Amendments and Responses to the Editor’s and Reviewers’ Comments

We are grateful to you for allowing us to revise and resubmit the paper. We would also like to thank the anonymous reviewer for his/her insightful and constructive comments. We have fully revised the paper in line with all your recommendations, which have helped us in further improving the quality of the current version of the paper. We have outlined below the specific amendments that we have carried out in response to each of your suggestions. We hope that we have succeeded in addressing the concerns raised in your comments.

Below, we detail our point-by-point responses to your comments (in bold and italic fonts). Our response starts with ***. All revisions are highlighted in the manuscript.

Reviewer-2Comments:

Comments and Suggestions for Authors

The subject of the article is to analyze the impact of environmental regulations and CSR disclosure on ecological innovations in enterprises.

Analysis based on data from 655 companies listed on stock exchanges in China in 2010-2020. The linear regression method was used in the data analysis.The subject of the article is important due to the need to protect the environment. Remarks:

  1. No purpose of the article, no research method in the abstract.

***We agree with the comment and have added an additional note of the purpose and research method of the articleat the beginning of the abstract (line 13-19):“The resource and environmental constraints on China's economic development become more prominent, thus there is an urgent need for enterprises to achieve green innovation  transformation to promote high-quality economic development. Based on Chinese A-share listed companies in Shanghai and Shenzhen Stock Exchange from 2010 to 2020, we obtained data on 655 listed companies in China from 2010-2020, a total of 7205 samples., and this study explores the influencing mechanism of environmental regulation on corporate green innovation and the moderating mechanism of CSR disclosure by constructing a nonlinear fixed-effect regression model.”

  1. The ESG acronym mentioned in the keywords is not developed and nowhere else is used.

***We apologize for the mistakes in the manuscriptandremove "ESG" from the keywords (line 29).

  1. In the article, there is no justification for the selection of a sample of companies, no justification for the selection of the data analysis method (linear regression), no justification for the selection of variables.

***We are thankful for this constructive comment. First, we have added the justification for the selection of a sample of companies (line303-308):“The main reasons for the selection of the time range are as follows. On the one hand, the data for measuring CSR based on the perspective of green innovation in this paper mainly comes from Hexun.com CSR rating database, which has continuously rated A-share listed companies since 2010; on the other hand, it lies in the fact that the concept of green development was gradually strengthened in China around 2012.”

Moreover,we have added the justification for the selection of the data analysis method (line 401-412):“The detailed reasons are as follows: (1) Fixed effect is a method that is suitable for comparing differences between variables and its interactions in research results. Essentially, it is also a method of controlling variables,which can prevent endogeneity problems associated with omitted variables and improves the accuracy of the estimated coefficients; (2) Time and industry fixed effects are designed to control for characteristics that are unique and do not vary with individuals at the time and industry levels respectively; (3) Environmental regulation is a typical game process between government and enterprises, the level of green innovation of enterprises varies with the in-tensity of environmental regulation. In order to test whether there is a non-linear relationship between the intensity of environmental regulation and green innovation of enterprises, this study introduces the quadratic term of the intensity of environmental regulation into the regression model.”

Finally,we have added the justification for the selection of variables:

“GI: Referring to Yu, Zhang and Bi [64]” (line 328)

“Control Variable: referring to Yu Zhang and Bi [64] and Zhang [68]” (line 368)

4.What does it mean that the hypothesis has been verified? Positive or negative?The H4 hypothesis is not formulated and is being verified.

***First, weare grateful for the comment,and wehave added a table of hypothesis testing results to the conclusions section (line 620):

Table 6. Results of the hypothesis test

Hypothesis

Significance

Impact

H1(a):There is a U-shaped curve relationship between the intensity of environmental regulation and enterprises green innovation

Significant

U

H1(b):Environmental regulation with a lag period has a more significant impact on green innovation

Significant

U

H2:CSR disclosure has a positive regulatory effect on the relationship be-tween environmental regulation and enterprise green innovation

Significant

positive

H3:Environmental regulation and CSR disclosure has a greater impact on the green innovation of non-state-owned enterprises.

Significant

positive

In Addition, we apologize for the writing mistake in the original manuscript and replace "H4" with "H3"(line 571).

Reviewer 3 Report

This manuscript analyzes the relationship between environmental regulation and green innovation, and the regulatory effect of CSR on 7,205 Chinese companies listed on the Shenzhen Stock Exchange and the Shanghai Stock Exchange between 2010 and 2021, making important contributions scientific literature in the field of sustainability. Therefore, first of all, I congratulate the authors for the investigative effort made and for the scientific contributions made.

Next, I make an assessment of the different sections, providing some considerations that help the authors to improve the original version of the manuscript.

In the first place, the authors carried out an extensive review of the scientific literature in the line of research, demonstrating that there is a high level of interest and that it is currently a relevant topic, even more so in emerging economies. However, I recommend reading some works in the recent literature that make relevant contributions considering the keywords used:

(a)   Corporate social responsibility and sustainability. A bibliometric analysis of their interrelationships.

(b)  Circular economy, growth and green growth as pathways for research on sustainable development goals: A global analysis and future agenda.

(c)   The impact of external institutional drivers and internal strategy on environmental performance.

(d)  Critical success factors and green supply chain management proactivity: shedding light on the human aspects of this relationship based on cases from the Brazilian industry.

(e)   A comprehensive decision making model for the evaluation of green operations initiatives.

For its part, the research hypotheses were based on previous findings, and were adequately exposed.

In the methods section, the sample was correctly described, as well as the exclusion criteria to have a broad and representative sample. The time horizon is broad and up-to-date, so it shows the current situation of the Chinese companies under study. The variables used are widely described and justified, as well as the corrections made to avoid bias. Additionally, a summary figure is included to collect all the information concisely. Finally, the reasoning for the selection of the regression method used is presented, introducing the moderating variable and the control variable, also presenting the descriptive statistics and the correlations between the variables used in the model.

In the results section, the authors adequately express the results obtained: they represent graphically and argue in the text the main results obtained. Here it would be relevant to present a table at the end of the section that summarizes how the research hypotheses were finally accepted or rejected. This would greatly help readers to get a quick overview of the main results obtained.

Finally, the conclusions section could be improved. The authors adequately express the conclusions of their research results and establish some recommendations that are interesting to improve the levels of adoption of environmental regulation and the development of green innovation. However, it would be interesting to expand this section by describing how the proposed research work has contributed to expanding and improving the scientific knowledge available to date.

In general terms, the authors carried out an important research effort, posing relevant and current research questions, obtaining results that contribute to improving the knowledge available so far on environmental regulation and the adoption of green innovation in emerging economies. The questions I raise should not detract from the effort, and are only intended to help improve the original version of the manuscript.

My best wishes.

Author Response

Dear editorsand reviewers:

A Report of Amendments and Responses to the Editor’s and Reviewers’ Comments

We are grateful to you for allowing us to revise and resubmit the paper. We would also like to thank the anonymous reviewer for his/her insightful and constructive comments. We have fully revised the paper in line with all your recommendations, which have helped us in further improving the quality of the current version of the paper. We have outlined below the specific amendments that we have carried out in response to each of your suggestions. We hope that we have succeeded in addressing the concerns raised in your comments.

Below, we detail our point-by-point responses to your comments (in bold and italic fonts). Our response starts with ***. All revisions are highlighted in the manuscript.

Reviewer-3Comments:

Comments and Suggestions for Authors

This manuscript analyzes the relationship between environmental regulation and green innovation, and the regulatory effect of CSR on 7,205 Chinese companies listed on the Shenzhen Stock Exchange and the Shanghai Stock Exchange between 2010 and 2021, making important contributions scientific literature in the field of sustainability. Therefore, first of all, I congratulate the authors for the investigative effort made and for the scientific contributions made.

Next, I make an assessment of the different sections, providing some considerations that help the authors to improve the original version of the manuscript.

  1. In the first place, the authors carried out an extensive review of the scientific literature in the line of research, demonstrating that there is a high level of interest and that it is currently a relevant topic, even more so in emerging economies. However, I recommend reading some works in the recent literature that make relevant contributions considering the keywords used:

(a)  Corporate social responsibility and sustainability. A bibliometric analysis of their interrelationships.

(b)  Circular economy, growth and green growth as pathways for research on sustainable development goals: A global analysis and future agenda.

(c)  The impact of external institutional drivers and internal strategy on environmental performance.

(d)  Critical success factors and green supply chain management proactivity: shedding light on the human aspects of this relationship based on cases from the Brazilian industry.

(e)   A comprehensive decision making model for the evaluation of green operations initiatives.For its part, the research hypotheses were based on previous findings, and were adequately exposed.

***We are thankful for this constructive comment and have added literature reviews on several of these topics:First,“Mavragani et al. [17] examined the extent to which the openness of market economy and the quality of the institution affect environmental performance, showing the Envi-ronmental Performance Index is positively correlated to institutional indicators. ”(line 90-93)

Second,“Thanh Nguyet Phan et al. [4] found that a more comprehensive environmental management system enables organizations to cope with the institutional pressures they face and thus improve environmental performance.” (line 93-95)

Third,“MeseguerSánchez Víctor, et al. [40] used bibliometric techniques and evaluated the contribution of current academic re-search to advancement of sustainable development agenda as expressed in the UN sustainable development goals targets”(line 152-154)

  1. In the methods section, the sample was correctly described, as well as the exclusion criteria to have a broad and representative sample. The time horizon is broad and up-to-date, so it shows the current situation of the Chinese companies under study. The variables used are widely described and justified, as well as the corrections made to avoid bias. Additionally, a summary figure is included to collect all the information concisely. Finally, the reasoning for the selection of the regression method used is presented, introducing the moderating variable and the control variable, also presenting the descriptive statistics and the correlations between the variables used in the model.

***We are very grateful to you for these positive comments on the academic and have added the reasoning for the selection of the regression method (line 401-412):“The detailed reasons are as follows: (1) Fixed effect is a method that is suitable for comparing differences between variables and its interactions in research results. Essentially, it is also a method of controlling variables ,which can prevent endogeneity problems associated with omitted variables and improves the accuracy of the estimated coefficients; (2) Time and industry fixed effects are designed to control for characteristics that are unique and do not vary with individuals at the time and industry levels respectively; (3) Environmental regulation is a typical game process between government and enterprises, the level of green innovation of enterprises varies with the intensity of environmental regulation. In order to test whether there is a non-linear relationship between the intensity of environmental regulation and green innovation of enterprises, this study introduces the quadratic term of the intensity of environmental regulation into the regression model.”

3.In the results section, the authors adequately express the results obtained: they represent graphically and argue in the text the main results obtained. Here it would be relevant to present a table at the end of the section that summarizes how the research hypotheses were finally accepted or rejected. This would greatly help readers to get a quick overview of the main results obtained.  

***We agree with the comment and have added a table at the results section that summarizes how the research hypotheses were finally accepted or rejected (line 620):

Table 6. Results of the hypothesis test

Hypothesis

Significance

Impact

H1(a):There is a U-shaped curve relationship between the intensity of environmental regulation and enterprises green innovation

Significant

U

H1(b):Environmental regulation with a lag period has a more significant impact on green innovation

Significant

U

H2:CSR disclosure has a positive regulatory effect on the relationship be-tween environmental regulation and enterprise green innovation

Significant

positive

H3:Environmental regulation and CSR disclosure has a greater impact on the green innovation of non-state-owned enterprises.

Significant

positive

  1. Finally, the conclusions section could be improved. The authors adequately express the conclusions of their research results and establish some recommendations that are interesting to improve the levels of adoption of environmental regulation and the development of green innovation. However, it would be interesting to expand this section by describing how the proposed research work has contributed to expanding and improving the scientific knowledge available to date. In general terms, the authors carried out an important research effort, posing relevant and current research questions, obtaining results that contribute to improving the knowledge available so far on environmental regulation and the adoption of green innovation in emerging economies. The questions I raise should not detract from the effort, and are only intended to help improve the original version of the manuscript.

*** We are thankful for this constructive comment and have added to the marginal contribution (line 622-641):

“The marginal contribution of this paper may be reflected in the following aspects. At the theoretical level, on the one hand, it provides evidence for the debate on whether the relationship between environmental regulation and enterprise green innovation is “linear” or “nonlinear”. At present, most researches on the economic consequences of environmental regulation focus on regional [24] and industry [11] levels. This paper conducts an in-depth investigation from the perspective of enterprises and finds that weak environmental regulation has a “cost effect” on enterprise green innovation, while strict environmental regulation has an “innovation compensation effect” on enterprise green innovation. And this study enriches the Porter’s hypothesis, environmental regulation and Institutional Pressure Theories. On the other hand, this paper enriches Schumpeter's innovation theory by taking green innovation as the break-through point. From the perspective of internal and external incentives, this paper at-tempts to use the resource-based theory and competitive advantage theory to interpret the mechanism of social responsibility information disclosure on environmental regulation and enterprise green innovation. We found that the disclosure of social responsibility information can significantly strengthen the relationship between environmental regulation and corporate green innovation, which not only can deepen the research on the mechanism of social responsibility information disclosure and the pre-drivers of corporate green innovation, but also can expand the research perspective of stakeholder theory.”

“At the practical level, the conclusions obtained can provide certain theoretical guidance and practical basis for governments at all levels to optimize and adjust the intensity of environmental regulation, improve environmental regulation policies and scientifically formulate governance measures, for enterprises to improve the level of social responsibility information disclosure and achieve ecological protection and high-quality development.”

We have also added limitations and future research (line 686-702):

 “As with most empirical research, our study had three limitations that provide potential directions for future research. First, in order to measure the intensity of environmental regulation, our study constructed a comprehensive evaluation and measurement system with the help of Z-Score standardization and entropy method, which can weaken the influence of measurement error to a certain extent. However, with the improvement and development of environmental regulation policies, its measurement method is multi-dimensional and dynamic. For example, commandandcontrol, market-incentive and public-participation environmental regulation can be classified according to the effect and degree of influence on corporate behavior and decision-making. Therefore, in the environment of extensive empowerment of social responsibility information disclosure, how the incentive mechanism for green innovation of different types of environmental regulations needs to be further studied.”

“Second, considering that the performance of corporate green innovation is closely related to factors such as digital strategic orientation and executive environmental awareness, how to study its possible moderating effects from the perspective of digital investment and executive characteristics still needs to be further tested.”

“Third, our study examines the differential impact of environmental regulation on green innovation driven by environmental regulation under the heterogeneity of enterprise property rights. Considering the correlation between the growth stage and marketization degree of enterprises and the performance of green innovation, the re-search on how to identify the differentiated impact of environmental regulation and enterprise green innovation based on different growth stages and marketization degree needs to be further enriched.”
